# Progress in the Pathogenesis and Treatment of Neuropsychiatric Systemic Lupus Erythematosus

**DOI:** 10.3390/jcm11174955

**Published:** 2022-08-24

**Authors:** Minhui Wang, Ziqian Wang, Shangzhu Zhang, Yang Wu, Li Zhang, Jiuliang Zhao, Qian Wang, Xinping Tian, Mengtao Li, Xiaofeng Zeng

**Affiliations:** Department of Rheumatology and Clinical Immunology, Chinese Academy of Medical Sciences and Peking Union Medical College, National Clinical Research Center for Dermatologic and Immunologic Diseases (NCRC-DID), Ministry of Science and Technology, State Key Laboratory of Complex Severe and Rare Diseases, Peking Union Medical College Hospital (PUMCH), Key Laboratory of Rheumatology and Clinical Immunology, Ministry of Education, Beijing 100730, China

**Keywords:** systemic lupus erythematosus, neuropsychiatric lupus erythematosus, pathogenesis, management, novel targeted therapies

## Abstract

Neuropsychiatric systemic lupus erythematosus (NPSLE) has a broad spectrum of subtypes with diverse severities and prognoses. Ischemic and inflammatory mechanisms, including autoantibodies and cytokine-mediated pathological processes, are key components of the pathogenesis of NPSLE. Additional brain-intrinsic elements (such as the brain barrier and resident microglia) are also important facilitators of NPSLE. An improving understanding of NPSLE may provide further options for managing this disease. The attenuation of neuropsychiatric disease in mouse models demonstrates the potential for novel targeted therapies. Conventional therapeutic algorithms include symptomatic, anti-thrombotic, and immunosuppressive agents that are only supported by observational cohort studies, therefore performing controlled clinical trials to guide further management is essential and urgent. In this review, we aimed to present the latest pathogenetic mechanisms of NPSLE and discuss the progress in its management.

## 1. Introduction

Systemic lupus erythematosus (SLE) is an autoimmune disease that may affect almost every organ [1,2]. SLE with nervous system involvement is known as neuropsychiatric systemic lupus erythematosus (NPSLE). NPSLE is a major contributor to morbidity and mortality in SLE patients. The American College of Rheumatology (ACR) defined 19 neuropsychiatric syndromes, ranging from central neurologic and psychiatric disorders to peripheral neuropathy [3,4,5]. The challenge now is that the underlying pathogenesis remains ambiguous [6,7,8], due to the limited access to nerve tissue, the complex nature of clinical manifestations, and overlap with non-lupus-associated neuropsychiatric events. These difficulties limit the optimization of NPSLE management.

In this review, we discuss the latest pathogenic mechanisms of NPSLE and explore new ideas and directions for the management of this complicated disease.

## 2. Pathogenesis of NPSLE

The exact immunopathogenesis of NPSLE is complex and unclear. Ischemic and autoimmune-mediated neuroinflammatory pathways are now considered two main, and probably complementary, pathogenetic mechanisms leading to NPSLE (Figure 1).

### 2.1. Ischemic Pathway

Ischemic injury to large- and small- blood vessels, mediated by antiphospholipid (aPL) antibodies, immune complexes, and complement activation leads to focal (e.g., stroke) and diffuse (e.g., cognitive dysfunction) neuropsychiatric events. Among these, aPL antibodies play a predominant role in the intravascular thrombosis [9]. Some studies have reported that SLE patients positive for aPL antibodies are approximately twice as likely to develop NPSLE than aPL-negative patients. aPL antibodies may also increase the risk of subclinical atherosclerosis, leading to a propensity for cerebral ischemia. The central nervous system is more susceptible than most tissues to thrombus formation, which accounts for the increased risk of stroke and transient ischemic attack seen in aPL antibody-positive patients [10]. Apart from thrombosis, aPL antibody positivity has also been correlated with other NPSLE manifestations, such as seizures, chorea, cognitive dysfunction, and myelopathy [11,12,13], especially psychosis [14,15,16]. Recent evidence suggests that aPL antibodies are also linked to direct neuronal damage by inducing oxidative stress and damage to neuronal cell membranes via the β2-glycoprotein. In an in vitro study, aPL antibodies bound to neurons and other CNS cells, and the intracerebroventricular injection of aPL induced a hyperactive behavior in animal models [17], thereby supporting a direct effect of these antibodies on the brain.

The aPL-mediated procoagulant state has traditionally been considered noninflammatory. However, a recent study found that mice deficient in C3 and C5 complement components were resistant to aPL-induced thrombosis and endothelial activation [18]. Thus, complement activation is associated with focal NPSLE, psychosis, and cognitive dysfunction, suggesting an additional inflammatory pathogenic component in NPSLE [19].

### 2.2. Neuroinflammatory Pathway

Autoimmune-mediated neuroinflammatory pathways with complement activation, enhanced the permeability of the blood–brain barrier (BBB), the intrathecal migration of neuronal autoantibodies, and the local production of pro-inflammatory cytokines, and other inflammatory mediators are associated with mostly diffuse neuropsychiatric manifestations, such as psychosis, mood disorders, and cognitive dysfunction [8,18,20,21].

#### 2.2.1. Enhanced Permeability of Brain Barrier

BBB disruption was the first pathophysiological mechanism proposed to play an imperative role in the development of NPSLE [22]. It establishes a structural and functional interface between the brain and general circulation to prevent the passive transfer of immune mediators from the blood to the central nervous system (CNS). Excessive levels of neurotransmitters, cytokines, chemokines, and peripheral hormones may influence BBB permeability [23].

Moreover, animal models of NPSLE have shown that increased BBB permeability is essential for autoantibodies to enter the brain [20,21] and then bind to neurons, which may lead to apoptosis [24]. However, the evidence for persistent BBB dysfunction is controversial [25].

Aside from BBB, the blood–cerebrospinal fluid barrier (BCSFB)—located at choroid plexus epithelial cells—is the natural ‘dam’ between the systemic circulation and the cerebrospinal fluid (CSF). It is a secretory epithelial structure surrounding a highly vascularized capillary plexus that produces cerebrospinal fluid (CSF) [26]. An increasing number of studies have focused on BCSFB in animal models, demonstrating that the choroid plexus epithelium has been identified as a route of entry into the CSF for pathogenic autoantibodies and leukocytes and as a primary site of neuropathology [27,28,29]. Additionally, some studies have implicated that in the absence of BBB dysfunction, the BCSFB could still be disrupted, supporting BCSFB dysfunction as a possible causative factor for immune mediators penetrating the brain [28]. 

Furthermore, the meningeal barrier and glymphatic system have also been proposed as potential sites of neuroimmune interactions, but their exact pathogenic roles await further validation in future studies [30,31]. 

#### 2.2.2. Autoantibody-Induced Inflammation

A typical feature of SLE is the formation of various autoantibodies, several of which are involved in NPSLE development. Here we will illustrate the identified autoantibodies that have been linked to NPSLE and their potential role in its pathogenesis. 

##### Anti-NMDAR Antibodies

N-methyl D-aspartate receptors (NMDARs) are receptors for the neurotransmitter glutamate, which is a major excitatory neurotransmitter that is important for many brain functions [32]. It has been reported that anti-NMDAR antibodies are related to the psychiatric manifestations of NPSLE [33,34,35]. 

Anti-NMDAR antibodies became important upon the observation that some anti-DNA antibodies might cross-react with NMDARs subunits on neurons [36]. These cross-reactive anti-NMDAR antibodies occur in SLE patients and are frequently associated with NPSLE [37,38,39]. 

The CSF titers of these antibodies are higher in patients with active diffuse NPSLE than in those with focal NPSLE or non-inflammatory CNS diseases [34,40]. In vitro studies have shown that anti-NMDARs may damage the BBB and penetrate the CNS [41]. Furthermore, the effect of anti-NMDARs is dose-dependent, as at low concentrations they seem to impair synaptic transmission, whereas at high concentrations they may cause neuronal apoptosis [32,33]. 

Nevertheless, these antibodies may also be present in SLE patients without neuropsychiatric involvement [42,43]. Thus, further research is needed to investigate the effect of anti-NMDARs on the pathogenesis of NPSLE development. 

##### Anti-RP Antibodies

Antibodies targeting the ubiquitous ribosomal P (RP) proteins have been associated with NPSLE, especially when manifested as psychosis and depression [42,44,45]. Anti-RP antibodies were predominantly detected in patients with SLE and were not detected in the control population [21]. The levels of anti-RP were higher in the serum/CSF of NPSLE patients with psychosis, depression, and asymptomatic cranial involvement [46], suggesting a potential role of anti-RP in the pathogenesis of NPSLE. Despite the above findings, some clinical studies that examined whether serum anti-RP antibodies correlated with psychosis have yielded inconsistent results [22]. Moreover, serum anti-RP antibodies are significantly associated with a worse prognosis in patients with diffuse NPSLE [47]. 

Importantly, the injection of anti-RP antibodies through the nervous system or peripheral circulation leads to cognitive impairment and depression in mice [21,48]. In vitro studies have shown that anti-RP antibodies could induce concentration-dependent neuronal dysfunction or apoptosis by increasing intracellular calcium release and disrupting protein synthesis [9,49].

Above all, anti-RP antibodies may be a relatively strong marker associated with psychiatric NPSLE.

##### AECAs

Anti-endothelial cell antibodies (AECAs) mediate the expression of adhesion molecules in endothelial cells. They are found in more than half of patients with NPSLE and are also correlated with psychosis and depression manifestations [50,51]. The activation of endothelial cells by AECAs might contribute to cerebral vasculopathy, which, in turn, induces the neuropsychiatric symptoms of SLE [8]. 

##### Anti-Ganglioside Antibodies

Gangliosides, spread across neurons’ surfaces, are crucial for signal transition [52]. One study reported that positivity for anti-ganglioside antibodies is frequent in lupus patients with PNS involvement [53]. However, this finding needs further investigation to achieve a consistent result [54]. 

Endothelial cells connected by tight junctions form the blood–brain barrier (BBB). After the BBB is compromised, antibodies gain access to the CSF while activated endothelial cells secrete pro-inflammatory cytokines, including interleukin-6 (IL-6) and interleukin-8 (IL-8). The associated signaling pathways involve the cytokine tumor necrosis factor-like weak inducer of apoptosis (TWEAK) to promote BBB disruption through the induction of inflammatory cytokines. Cytokines and chemokines, such as IL-6 and a proliferation-inducing ligand (APRIL), enhance B-cell activation and survival.

Immune complexes could induce interferon-α (IFN-α) production. IFN-α could activates microglial engulfment of neurons and directly damages them. Microglial activation further propagates local cytokine and chemokine signaling cascades. Furthermore, IFN-α enhances microglial cytokine and chemokine (IL-6, IL-8, MCP-1, IP-10) production. Finally, several neuropathic autoantibodies have been implicated in NPSLE. Autoantibodies, such as anti-NMDAR and anti-RP, directly bind to neurons and lead to neuronal dysfunction or apoptosis. Following neuronal cell damage, antibodies form immune complexes with neuronal antigens, contributing to the diffuse neuronal damage/dysfunction in the brain. Created with biorender.com.

#### 2.2.3. Cytokines-Mediated Inflammation

In CNS, cytokines are expressed at low levels by neurons, astrocytes, microglia, and oligodendrocytes. The expression of genes encoding cytokines and their receptors in the brain suggests that cytokines contribute to the normal physiological functions of CNS. Cytokines and other immune factors are important for the modulation of brain development and affect adult neuronal plasticity, leading to cognitive and mood disorders [55]. Below, we will discuss the cytokines that potentially participate in the pathogenesis of NPSLE.

##### TWEAK/Fn14

The tumor necrosis factor-like weak inducer of apoptosis (TWEAK), a member of the tumor necrosis factor (TNF) superfamily of cytokines, promotes the activation of NF-kB and mitogen-activated protein kinase (MAPK) by binding to fibroblast growth factor-inducible 14 (Fn14), a 14 kDa member of the TNF receptor superfamily. Fn14 is expressed in a variety of cells and tissue types, including fibroblasts, endothelial cells, and epithelial cells [56].

TWEAK plays an important role in BBB disruption and the development of NPSLE [57]. Fn14 exhibited upregulation within the cerebral cortex of lupus-prone mice. In addition, the severe depression-like behavior observed in MRL/lpr mice was significantly reduced in Fn14-deficient mice, indicating that Fn14 improved depression and cognitive function [58]. Moreover, the intracerebroventricular injection of TWEAK in wild-type mice induced cognitive dysfunction and depression-like behavior through increased BBB permeability and accelerated neuronal apoptosis [59]. However, this cytokine seems to be elevated in the CSF of SLE patients, regardless of the presence or absence of neuropsychiatric symptoms. 

##### IL-6

Interleukin-6 (IL-6) is thought to have the strongest positive association with NPSLE [60]. The elevated intrathecal levels of IL-6 have been found in patients with diffuse NPSLE, such as those experiencing an acute confusional state or psychosis [61]. In addition, the positive correlation between IL-6 levels and the levels of the neuronal degradation product denominated neurofilament light chains (NFL), which indicates that IL-6 exerts destructive effect on nerve cells [62]. Nevertheless, research on the correlation between serum IL-6 and psychiatric NPSLE provided inconclusive results [63]. This difference needs to be further explored.

##### IFN-α

An animal models have demonstrated a significant association between IFN-α in the CSF and NPSLE, identifying a novel IFN-α-dependent mechanism for NPSLE. IFN-α has been proposed to cause damage by activating microglia in the CSF and stimulating the microglial engulfment of neuronal cells [64]. IFN-α may also impair brain function by altering the levels of neurotransmitters and generating damage by the secondary release of cytokines and chemokines, such as IL-6- and interferon-gamma-inducible protein-10 (IP-10) [65].

Additionally, neuropsychiatric manifestations observed in lupus-prone animal models were reversible with IFN-α inhibition, indicating that IFN-α is imperative in the pathogenesis of NPSLE [64].

##### BAFF and APRIL

The TNF family ligands B-cell activating factor of the TNF family (BAFF) and APRIL are crucial in the survival, differentiation, and isotype switching of B lymphocytes [66].

One study found a close relationship between APRIL in the CSF and NPSLE but not between BAFF in the CSF and NPSLE [67]. To date, there have been few studies regarding their exact role in the pathogenesis of NPSLE. 

#### 2.2.4. Brain-Resident and Infiltrating Cells

Apart from structural changes, autoantibodies and cytokines, alterations in brain-resident cells in the CNS may be instrumental in the development of NPSLE. 

Microglia, the resident macrophage cells of the brain, are the main antigen-presenting cells (APCs) in the CNS. They play a fundamental role in regulating BBB function and shaping the brain circuits. They could also secrete various cytokines, chemokines, prostaglandins, and reactive oxygen species [68].

Increasing evidence supports an active role for microglial cells in the pathogenesis of NPSLE. Activated microglia are a feature of several models of the lupus-prone mouse [69,70]. MLR/lpr mice lacking estrogen receptor alpha experienced a significant neuropsychiatric disorder, which correlated with a decreased number of activated microglial cells and an accompanying reduction in CNS inflammation [71]. 

The intrathecal synthesis of cytokines as a potential mechanism of damage, and neural damage may develop in NPSLE without the involvement of factors derived from blood [72]. Although the cellular origin of these cytokines production in the brain remains unknown, macrophages and endothelial cells (ECs), as well as brain-derived microglia and astrocytes, are probable sources of these cytokines. Microglial depletion by colony-stimulating factor-1 receptor (CSF1R) inhibitors resulted in preserved neuronal integrity in an inducible mouse model (NMDAR peptide immunized BALB/c mice). Interestingly, another study showed that the administration of captopril (an angiotensin-converting enzyme (ACE)) inhibitor significantly reversed the activation of microglia and improved the cognitive function of mice [73]. 

Large clusters of leukocytes infiltrate the choroid plexus in vitro. The analysis of the choroid plexus indicated a tertiary lymphoid structure formation, with evidence of APC-lymphocyte interactions, cytokine production, and in situ somatic hypermutation [74]. 

## 3. Current Management of NPSLE

The management of NPSLE can be challenging, because of the complexity of its pathogenesis, difficulty in its accurate diagnosis, and a lack of clinical trials in NPSLE. Current treatment options for NPSLE are usually derived from observational studies and refer to the experience of treatment of other SLE subtypes, such as lupus nephritis and similar neuropsychiatric disorders [8,75]. 

Initially, it is crucial to develop pragmatic therapeutic strategies to determine the attribution of nervous system disease to SLE, non-SLE causes, or both. Confounders and mimics should be ruled out and the symptoms should be initially attributable to SLE at the beginning. The goal of management of NPSLE is to meet two criteria. First, symptomatic therapy is necessary: anti-epileptics for seizures, and anxiolytics, antidepressants, mood-stabilizers, or antipsychotics should be administered as appropriate. Neurotrophic and neuroleptic agents were generally adopted in case with peripheral nervous system involvement [4]. The treatment of the underlying SLE process should be undertaken based on whether the pathogenesis is primarily related to an inflammatory or ischemic disease pathway (Figure 2).

### 3.1. Inflammatory Pathway Therapies

Glucocorticoids have been a cornerstone in the treatment of various manifestations of NPSLE, especially in those associated with an immune-inflammatory pathogenesis [76,77]. High-dose glucocorticoids, alone or in combination with cyclophosphamide, azathioprine, and mycophenolate mofetil, are reported to be effective, but their use is mainly based on patient’s clinical experience of disease severity and their clinician’s preference. Given the evidence linking glucocorticoids use to cumulative organ damage in SLE [78] and its associated psychiatric symptoms [79,80], alternative therapeutic strategies are essential. 

Unfortunately, high-level clinical evidence regarding the optimization of NPSLE treatment is lacking. Only two of these agents (oral prednisone and intravenous cyclophosphamide) have been subjected to clinical trials for NPSLE [76], and both had positive outcomes. In addition, a regimen of oral cyclophosphamide for 6 months followed by azathioprine maintenance therapy was effective for the treatment of lupus psychosis [81]. 

The examination of biological agents in NPSLE is limited to uncontrolled studies. Open studies on B-lymphocyte depletion with rituximab used alone or in combination with conventional immunosuppressive agents, including cyclophosphamide, have reported favorable results in children [82] and adults [83] with NPSLE; however, this requires further study. Perhaps of relevance is the observation that rituximab can be beneficial in other inflammatory neurological conditions, such as neuromyelitis optica, anti-NMDAR encephalitis, and opsoclonus–myoclonus syndrome [84,85]. Studies of belimumab suggested a beneficial response to belimumab in SLE, but patients with severe NPSLE were excluded from these clinical trials [86].

### 3.2. Ischemic Pathway Therapies

Cerebral ischemia attributed to NPSLE, such as transient ischemic attacks and stroke attributed to NPSLE events and is thought to be correlated with aPL antibodies. Thus, the primary prevention of cerebral ischemia in NPSLE is linked to a reduction in prothrombotic risk.

Low-dose aspirin is recommended for patients with cardiovascular risk factors [87]. However, a previous review of primary prevention in antiphospholipid syndrome (APS) concluded that the current evidence does not support either the use of low-dose aspirin or warfarin [88]. The optimal target international normalized ratio (INR) in such cases is inconsistent [87] and the recommended INR target in patients with APS is 2.5–3.0. In patients with recurrent thrombosis despite optimal warfarin therapy, the INR should be kept at 3.0–4.0. Nevertheless, there is no obvious difference between low-intensity (target INR 2.0–3.0] and high-intensity (target INR > 3.0) warfarin in the prevention of recurrent thrombosis in controlled trials with APS patients [89,90]. Therefore, well-designed clinical trials are needed to address this issue. Currently, the data are insufficient to recommend the use of direct novel oral anti-coagulants to prevent aPL antibody-mediated thromboembolic events [91].

Potential adjunctive therapies, especially in patients with arterial thrombosis and recurrent venous thrombosis, include antimalarials and statins [87]. Statins can prevent endothelial cell activation secondary to aPL antibodies [92], while antimalarial agents are protective against thrombosis in patients with SLE [93].

## 4. Promising Targeted Therapies

Due to the lack of understanding of the exact pathogenic mechanisms behind this condition as well as its diverse neuropsychiatric manifestations, we have limited experience in targeted therapies for patients with NPSLE. The attenuation of neuropsychiatric diseases in related animal models demonstrates the potential for targeted therapies, which are based on a current understanding of the pathogenesis of NPSLE (Table 1). 

### 4.1. Complement Inhibitors

A study devoted to the role of the complement protein C5a in the brain vasculature indicated that the C5a/C5aR signaling plays a key role in disrupting the BBB integrity [94]. C5aR was blocked the complement cascades and retained their protective functions, which relieved the symptoms of NPSLE. Thus, C5aR may be a potentially important therapeutic target for NPSLE [95]. In addition, the presence of C5b-9 deposits in most patients with NPSLE is important in the interaction between circulating autoantibodies and thrombo-ischemic lesions observed in NPSLE. Therefore, the complement inhibitor eculizumab may have novel therapeutic potential for NPSLE [68,96]. However, the efficacy of such treatment in NPSLE remains to be further investigated.

### 4.2. BBB-Targeted Therapies

BBB dysfunction exposes the brain to components of the blood that are normally excluded. Decreasing the permeability of the BBB could be advantageous for a patient with NPSLE. This could prevent autoantibodies, cytokines, and non-immune proteins from causing inflammation, neuronal hyperexcitability, and degeneration [97].

The restoration of normal BBB function is a potential therapeutic strategy. Two compounds that reduce BBB permeability (GW0742, a peroxisome proliferator-activated receptor β/δ agonist [98], and KD025, a Rho kinase inhibitor [99]) have been studied in experimental systems and may be considered as therapies. 

### 4.3. MMPs Inhibitors

Matrix metalloproteinases (MMPs) are proteolytic enzymes that could degrade basement membranes, disrupt inter-endothelial tight junctions, and activate membrane-bound proinflammatory molecules. Among those, MMP-9 induces the production of cytokines and leukocyte adhesion molecules by endothelial cells, facilitating the entry of leukocytes and proteins into the CSF [100,101]. 

Studies have demonstrated an association between CSF/serum levels of MMP-9, psychiatric NPSLE, and the markers of neuronal/astrocytic damage [102]. MMP-9 may contribute to the pathogenesis of psychiatric NPSLE by stimulating T-cell migration. Therefore, the inhibition of MMPs, especially MMP-9 [103], could introduce a novel biological agent and may be beneficial in NPSLE. 

### 4.4. IFN-α/β Receptor Antagonists

Anifrolumab, a type I interferon receptor antagonist that binds to the IFN-α/β receptor, has been successfully used in phase III clinical trials for SLE treatment [104]. The adoption of anifrolumab leads to a substantial reduction in moderate-to-severe active SLE; however, patients with severe NPSLE were not involved in these trials [105]. Therefore, the results may not support the efficacy of this drug in the treatment of NPSLE. However, as mentioned before, the type I interferon receptor inhibition decreases microglia related synaptic loss and attenuates anxiety-like behavior and cognitive deficits in lupus-prone mice [38]. This implies that type I interferon inhibition may be an option for the treatment of NPSLE in the future [106], especially in patients with a strong type I interferon signature. 

### 4.5. BTK Inhibitors

Bruton’s tyrosine kinase (BTK) is essential for B cell function, including B cell development and survival; for crystallizable fragment (Fc) receptor and toll-like receptor (TLR) signaling in macrophages; and for macrophage polarization [107,108,109].

The inhibition of this pathway using of a specific inhibitor (BI-BTK-1) in MRL/lpr mice resulted in the decreased infiltration of macrophages, T cells, and B cells in the choroid plexus and improved cognitive function [110]. 

Ibrutinib, a selective BTK inhibitor, could potentially prove useful in the treatment of neuropsychiatric disease, such as SLE [111]. This inhibitor has already have already been approved for clinical use in hematological indications [112] and results from ongoing early phase clinical trials of BTK inhibitors in patients with SLE are eagerly awaited. 

### 4.6. S1P Receptor Modulator

Fingolimod, a sphingosine-1-phosphate (S1P) receptor modulator, was shown to decrease macrophage infiltration and proinflammatory cytokine secretion by microglia, resulting in improved spatial memory and reduced depression-like behavior in MRL/lpr mice [113]. Fingolimod administration attenuated neuropsychiatric manifestations, reversed the entry of immune components, and decreased BBB leakage in the above studies [114].

In addition to the possible mechanisms mentioned above, fingolimod-treated microglia revealed the downregulation of multiple immune-mediated pathways, including NF-kB signaling and the IFN response, with the negative regulation of type I IFN-mediated signaling [113]. 

In line with the approved use of fingolimod in relapsing–remitting multiple sclerosis, these studies may support the potential use of fingolimod as a therapeutic strategy for NPSLE patients.

### 4.7. ACE Inhibitors

ACE inhibitors, such as captopril and perindopril, improve cognitive status and neuronal functions in lupus-prone mice [73]. 

Additionally, ACE inhibitors treatment in a lupus-prone model suppressed microglial activation, which in turn preserved dendritic complexity in hippocampal neurons. Further analysis is needed to explore the specific pathogenesis, but this therapeutic regimen may also be considered in the future to treat cognitive impairment in NPSLE patients. 

### 4.8. CSF1R Inhibitors

Macrophage colony stimulating factor 1 receptor (CSF1R) is an important regulator of both macrophage and microglial functions. It plays a pivotal role in macrophage and microglia development, survival, and activation [115]. 

In a mouse model, the inhibition of CSF1R signaling reduced the brain expression of pro-inflammatory cytokines and attenuated depression performance [116], indicating that CSF1R is a potential target for treating NPSLE in the future.

### 4.9. Nogo-a/NgR1 Antagonists

Neurite outgrowth inhibitor-A (Nogo-a) with its respective receptor, NgR1, forms a signaling pathway that mediates the inhibition of neuron generation. Patients with NPSLE have significantly increased levels of Nogo-a/NgR1 in the CSF, compared to other neurological diseases. It has also been demonstrated in MLR/lpr mice that the administration of Nogo-66 [117], an antagonist of Nogo-a, improved cognitive function, decreased the expression of pro-inflammatory components, and reduced axonal degeneration and demyelination, implying that Nogo-a is a potential therapeutic target for cognitive impairment in NPSLE.

### 4.10. JAK Inhibitors

JAK inhibitors, which interfere with the JAK-STAT signaling pathway, are small molecules that penetrate the BBB [118] and reduce the production of several cytokines, including type I IFNs. Tofacitinib, a JAK1/JAK3 inhibitor is currently in phase II studies for SLE treatment and is worthy of consideration in this regard [119]. However, whether it is effective for NPSLE still requires further research.

## 5. Conclusions

Neuropsychiatric events in SLE patients are common and tend to be heterogeneous, and many knowledge gaps remain in our basic understanding of NPSLE and its clinical management. Current available therapies are largely empirical, and most evidence is derived from studies in animal models, which do not manifest the full spectrum of human NPSLE. 

Advances remain to be made in enhancing the understanding of the pathogenesis, and optimizing our ability to diagnose, prognosticate, and treat NPSLE. Innovative strategies targeting the brain structural barrier, specifically autoantibodies, cytokines, and brain-resident cells, are worthy of exploration and further study. 

We anticipate that some of these pathways could serve as targets for the development of a new therapeutic strategies. Promising research efforts into novel targeted therapies and improved diagnostic tools are ongoing; however, much work remains to be done to optimize our ability to diagnose, prognosticate, and treat NPSLE. 

## Figures and Tables

**Figure 1 jcm-11-04955-f001:**
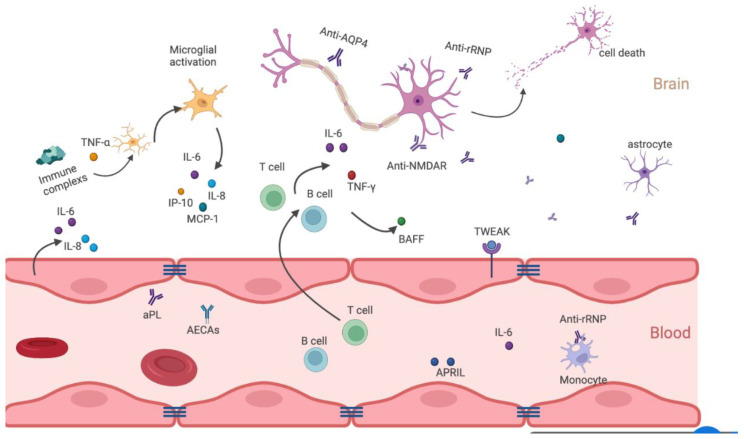
Pathogenetic mechanisms in diffuse NPSLE.

**Figure 2 jcm-11-04955-f002:**
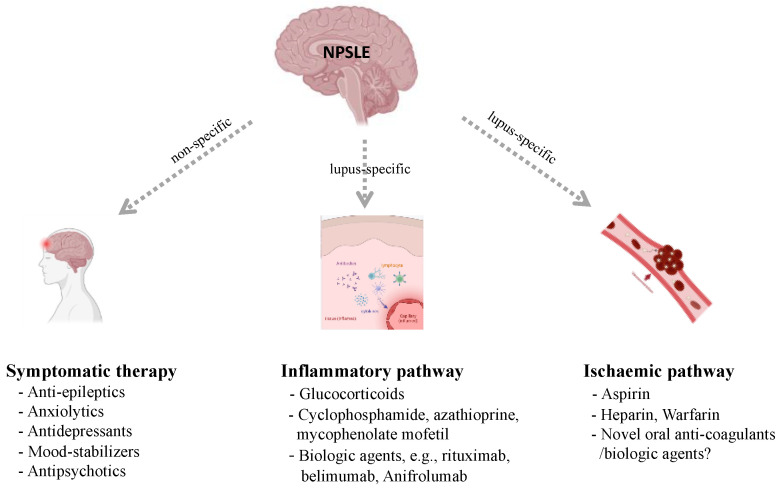
Management for patients with neuropsychiatric systemic lupus erythematosus (NPSLE). Created with biorender.com.

**Table 1 jcm-11-04955-t001:** Promising targeted therapies in NPSLE.

Promising Targeted Therapies	Underlying Mechanisms and Clinical Findings	Experimental Arrangement	Potential Drugs
Complement inhibitors	Complement signaling promotes the loss of BBB integrity. Blocking the complement cascades relieved the symptoms of NPSLE. Complement deposits were present in most of patients with NPSLE.	Human brain autopsies	Eculizumab
BBB-targeted therapy	BBB disruption is essential in the neuronal damage process. Restoration of normal BBB function may reduce the development of neuropsychiatric manifestations	Human and mouse cells; C57 BL/6J mice, respectively	GW0742, a peroxisome proliferator-activated receptor β/δ agonist; KD025, a rho kinase inhibitor.
MMPs inhibitors	There is an association between CSF/serum levels of MMP-9, psychiatric NPSLE, and markers for neuronal/astrocytic damage. MMP-9 may contribute to the pathogenesis of psychiatric NPSLE by stimulating T-cell migration	-	-
IFN-α/β receptor antagonists	IFN receptor inhibition decreased microglia-related synaptic loss and attenuated anxiety-like behavior and cognitive deficits in animal models.	564Igi lupus-prone mice	Anifrolumab
BTK inhibitors	Use of BI-BTK-1 (an inhibitor of BTK) in MRL/lpr mice, decreased the infiltration of macrophages, T cells, and B cells in the choroid plexus, and improved cognitive function.	MRL/lpr mice	Ibrutinib; Evobrutinib
S1P receptor modulator	S1P receptor modulators decreased proinflammatory cytokine secretion by microglia and significantly improved spatial memory and depression-like behavior. Fingolimod (a S1P receptor modulator) treatment attenuated neuropsychiatric manifestations, reversed the entry of immune components, and decreased BBB leakage. Fingolimod-treated microglia revealed down- regulated of multiple immune-mediated pathways, including NF-kB signaling and the IFN response with the negative regulation of type I IFN-mediated signaling.	MRL/lpr mice	Fingolimod
ACE inhibitors	ACE inhibitors treatment suppressed microglial activation and promoted cognitive status.	BALB/c mice	Captopril; Perindopril
CSF1R inhibitors	CSF1R is essential in both macrophage and microglia function. Inhibition of CSF1R signaling in MRL/lpr mice reduced the brain expression of proinflammatory cytokines and attenuated depression performance.	MRL/lpr mice	GW2580, a small CSF-1R kinase inhibitor; depletion of microglia
Nogo-a/NgR1 antagonists	Nogo-a/NgR1 in the CSF is significantly increased in NPSLE. Nogo-a/NgR1 antagonists improved cognitive function, decreased the expression of pro-inflammatory components, and reduced axonal degeneration and demyelination.	MRL/lpr mice	Nogo-66
JAK inhibitors	JAK inhibitors penetrate the BBB and reduce the production of several cytokines, including type I IFNs.	-	Tofacitinib

## Data Availability

Not applicable.

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
