# Peer review of "Progress in the Pathogenesis and Treatment of Neuropsychiatric Systemic Lupus Erythematosus"

_jcm, 2022, doi:10.3390/jcm11174955_

Round 1

Reviewer 1 Report

This is a review regarding what is known about the pathogenesis and treatment of neuropsychiatric lupus, which is surprisingly little. This being said, there are a number of important studies regarding the condition that are not mentioned, including that of known pathological studies. This, in addition to imprecise language within the paper regarding neuropsychiatric lupus, and the overall format of the paper make the review of little value to the clinical audience. 

The language of the paper requires significant revision, as many ideas appear to be incompletely expressed. There are innumerable typographical and format errors, as an example, reference 112 appears to be an error.

Author Response

Response to Reviewer 1 Comments

MANUSCRIPT ID: jcm-1837443

MANUSCRIPT TITLE: Progress in the pathogenesis and treatment of neuropsychiatric systemic lupus erythematosus

Dear Reviewer:

First of all, please let me express my gratitude for your hard work and efforts in my article.

We thank you for your helpful and constructive suggestions, which have enriched the manuscript and produced a better and more balanced account of the research. The manuscript has been rechecked and appropriate changes have been made in accordance with the reviewers’ suggestions using the “Track Changes” function by red-colored font.

The responses to their comments have been prepared and attached herewith/given below. We hope that the revised manuscript is now suitable for publication in your journal.

Point 1:

This is a review regarding what is known about the pathogenesis and treatment of neuropsychiatric lupus, which is surprisingly little. This being said, there are a number of important studies regarding the condition that are not mentioned, including that of known pathological studies.

Response 1: Thank you for your suggestion and questions. We have added the relevant information.

(In Pathogenesis of NPSLE section, Page 1-2.)

Point 2:

This, in addition to imprecise language within the paper regarding neuropsychiatric lupus, and the overall format of the paper make the review of little value to the clinical audience. 

Response 2:

Thank you for the suggested correction.

We apologize for the error and have modified the term “neuropsychiatric lupus (NPLE)” to “neuropsychiatric systemic lupus erythematosus (NPSLE)”.

We have also changed the overall format of paper to highlight the value of our findings.

(In Pathogenesis of NPSLE section, Page 1-6.)

Point 3:

The language of the paper requires significant revision, as many ideas appear to be incompletely expressed. There are innumerable typographical and format errors, as an example, reference 112 appears to be an error.

Extensive editing of English language and style required.

Response 3:

Thank you for your suggestion. Before resubmission, two persons proficient in written English have edited the manuscript. We also turned to a professional language editing service to help us further improve the manuscript. 

Reviewer 2 Report

It is a comprehensive description of neuropsychiatric SLE, and in my opinion the content is almost fine with a few exceptions.

This is not a content issue, but are the term "neuropsychiatric lupus" and its abbreviations "NP-LE" appropriate for terminology?

In the PubMed search results, the idiom "neuropsychiatric systemic lupus erythematosus" has 539 hits, but "neuropsychiatric lupus erythematosus" has 57 hits. Similarly, the abbreviation "NP-SLE" has 97 hits, but there is only one article with the abbreviation "NP-LE.

The term "Neuropsychiatric systemic lupus erythematosus (NP-SLE)" seems to be the appropriate one to use here.

Since this is a manuscript on NP-SLE, I understand that there is no need to discuss APS in detail, but the statement "anti-phospholipid (aPL) antibodies bind to several antigens" on page 3, line 14, is not accurate and should be replaced. (aPL does not directly recognize phospholipids.)

Minor point

On page 3, line 18, "microinfarction" appears twice in a row.
